# Virtual Reality Combined with Artificial Intelligence (VR-AI) Reduces Hot Flashes and Improves Psychological Well-Being in Women with Breast and Ovarian Cancer: A Pilot Study

**DOI:** 10.3390/healthcare10112261

**Published:** 2022-11-11

**Authors:** Danny Horesh, Shaked Kohavi, Limor Shilony-Nalaboff, Naomi Rudich, Danielle Greenman, Joseph S. Feuerstein, Muhammad Rashid Abbasi

**Affiliations:** 1Department of Psychology, Bar Ilan University, Ramat Gan 5290002, Israel; 2Department of Psychiatry, New York University Grossman School of Medicine, New York, NY 10016, USA; 3Bubble Ltd., Bnei Brak 4522567, Israel; 4Department of Medicine, Columbia University, New York, NY 10016, USA; 5Oncology & Hematology Specialists, P.A., Mountain Lakes, NJ 07046, USA

**Keywords:** artificial intelligence, breast cancer, CBT, hot flashes, virtual reality

## Abstract

Background and aims: Breast and ovarian cancers affect the lives of many women worldwide. Female cancer survivors often experience hot flashes, a subjective sensation of heat associated with objective signs of cutaneous vasodilatation and a subsequent drop in core temperature. Breast and Ovarian cancer patients also suffer from sleep difficulties and mental health issues. The present study aimed to assess the effectiveness of Bubble, a novel artificial intelligence–virtual reality (AI–VR) intervention for the treatment of hot flashes in female breast or ovarian cancer patients. Methods: Forty-two women with breast and/or ovarian cancer participated in the study. The mean age was 47 years (range: 25–60 years). Patients suffered from hot flashes at different frequencies. They used Bubble, a virtual reality (VR) mobile psychological intervention based on elements from both cognitive behavioral therapy and mindfulness-based stress reduction. The intervention took place in a VR environment, in a winter wonderland setting called Frosty. Patients were instructed to use Bubble at home twice a day (morning and evening) and when experiencing a hot flash. Participants were asked to use the application for 24 consecutive days. Before and after this 24-day period, patients completed self-report questionnaires assessing hot flashes, general psychiatric distress, perceived stress, illness perception, sleep quality, and quality of life. Results: Between pre- and post-intervention, participants reported a significant reduction in the daily frequency of hot flashes, stress, general psychiatric distress, several domains of QOL, and sleep difficulties, as well as an improvement in illness perception. In addition, they reported very high satisfaction with Bubble. Importantly, both age and baseline levels of psychopathology moderated the effect of Bubble on sleep difficulties. Discussion: This study showed preliminary evidence for the potential of VR interventions in alleviating hot flashes and accompanying mental distress among those coping with breast and ovarian cancer. VR is a powerful therapeutic tool, able to address mind–body aspects in a direct, vivid way. More studies are needed in order to fully understand the potential of this unique intervention.

## 1. Introduction 

Breast cancer is the most common cancer in women and the leading cause of death in women aged <55 years [1]. Among these women, 70–85% of breast cancer survivors experience hot flashes, an increase of about 20–35% compared with the general population [2]. Hot flashes are a subjective sensation of heat associated with objective signs of cutaneous vasodilatation and a subsequent drop in core temperature associated with sweating, flushing, palpitations, anxiety, panic, and irritability [3]. In addition, breast and ovarian cancer survivors often experience insomnia or interrupted sleep and reduced quality of life [4]. 

While endogenous estrogen can alleviate these vasomotor symptoms, concerns exist regarding its use in women with and without breast cancer. These concerns have led to extensive efforts to find well-tolerated and efficient non-hormonal therapies for women suffering from these troublesome symptoms. Postmenopausal women and breast cancer survivors often experience psychological difficulties, sleep impairments, and cognitive dysfunction beyond that attributable to vasomotor symptoms alone [5]. Therefore, an integrative intervention that addresses these multiple symptoms is undoubtedly warranted. 

Hunter and Liao [6] demonstrated that distressing or problematic hot flashes were predicted by depression, anxiety, and low self-image but not by the frequency of hot flashes. In addition, women with the same frequency of hot flashes can experience different intensities and diverse emotional responses to their hot flashes. Some may react with avoidance and depression, while others may feel little impact of hot flashes on their mood [7]. 

Integrative or complementary therapies including yoga, hypnosis, and relaxation training/paced respiration, as well as psychotherapeutic interventions such as mindfulness-based stress reduction and cognitive behavioral therapy (CBT) have been studied as effective treatments for stress, insomnia, depression, anxiety, pain, blood pressure regulation, and a myriad of other emotional and physical difficulties experienced by breast cancer patients [8]. CBT, with its emphasis on cultivating adaptive cognitive schemas and reducing maladaptive behavioral patterns such as avoidance, is considered the gold-standard treatment for many of the symptoms of menopause, including emotional distress and insomnia [9]. In the past two decades, third wave CBT interventions have also been widely studied, placing more emphasis on the acceptance of negative emotions and sensations, as well as on focusing attention on the present moment, in a nonreactive manner. Mindfulness-based interventions, which are part of third-wave CBT, have been shown to yield significant improvements in stress and behavioral symptoms, as well as in inflammation markers among breast cancer patients [1]. Relaxation training was also studied and was shown to reduce the incidence, severity, and distress of hot flashes in women with primary breast cancer compared with controls [2]. However, while the interventions above show promise, there is still much room for improvement in developing novel ways to treat women with breast cancer. The present study focuses on a virtual reality treatment, examining its potential for alleviating the physical and emotional distress of this population. 

Virtual reality (VR) is a computer technology that uses virtual reality headsets or multi-projected environments, sometimes in combination with physical environments or props, to generate realistic images, sounds and other sensations that simulate a user’s physical presence in a virtual or imaginary world. There are two main categories of VR, immersive and non-immersive. Non-immersive VR is where the user is connected to the virtual world but still has the possibility of communicating with the environment. The sense of presence can be increased in non-immersive VR by using a three-dimensional (3D) display. Presence, or immersion, is a variable that influences the attention of the user [10]. Full immersion uses a head mounted display, blocking the users’ view of the real world and presenting them with a total computer-generated environment completely separate from their reality and environment. This technology offers patients a safe environment and an opportunity to therapeutically intervene using the elements of mind–body medicine [11]. Furthermore, the confluence of VR solutions can be further enhanced by artificial intelligence (AI), particularly computer vision and natural language processing. AI is the study and implementation of techniques that allow actions requiring intelligence on the part of a human to be performed on computational devices. The purpose of AI is to capture the scarce resource of intelligence and then promote and use it through the spread of computer technology [12]. 

VR has been used extensively in treating individuals suffering from a variety of physical and psychological conditions, including pain, brain injuries, anxiety, and PTSD [13]. A review of 19 studies using VR therapies in cancer patients showed that 4 out of 8 studies found significant differences in pulse rate in the VR group compared with controls [14]. 

Studies of VR in women receiving chemotherapy [15], including older women (50 years+) with breast cancer [16] as well as adults with colon and lung cancer receiving chemotherapy [17] and children ages 10–17 receiving chemotherapy [18], all showed decreases in symptom distress (e.g., nausea, mood disturbance, insomnia, pain, mobility, fatigue), a perception that treatment times were shortened, and a belief that treatment with VR was better than chemotherapy alone. The majority of patients did not report any ‘cyber sickness’ (i.e., nausea and discomfort that can last for hours after participating in VR applications) and the intervention was generally found to be easy to implement. However, since VR is a relatively new technology, more studies are needed in order to understand its full potential in alleviating distress among cancer patients. In addition, while VR studies of cancer patients do exist, the vast majority are not based on technologies that were designed specifically to treat one core symptom of cancer (e.g., hot flashes, nausea), as well as the psychological symptoms which may accompany cancer. Finally, in this era of personalized medicine, it is crucial to ascertain which treatments fit which patient [19]. Thus, an increasing number of studies in the fields of medicine and psychotherapy are currently incorporating examinations of moderators of treatment response, including sociodemographic, medical, and psychological factors that may affect outcome. Such studies are still rather scarce in the field of VR therapy. 

To fill these gaps in research, we report on a new study using Bubble, a novel artificial intelligence–virtual reality (AI–VR) intervention for the treatment of hot flashes in female breast and/or ovarian cancer patients between the ages of 18 and 60. Our main aim was to assess the effectiveness of this intervention on the severity and frequency of hot flashes. We also set out to examine whether Bubble alleviates psychological symptoms often reported by those coping with cancer, including stress, anxiety, reduced sleep quality, negative illness perception/identity, and reduced quality of life. Finally, we aimed to assess whether VR technology assists certain patients more than others as a function of demographic and psychological background. Here, we focused on two variables that might moderate VR treatment response: one’s age and one’s level of psychological distress pre-treatment. These two factors were found to play a major role in previous psychotherapy and integrative medicine studies [20,21].

In this study, artificial intelligence was used to help personalize the experience using an AI-driven algorithm to make sure that every aspect of engagement with the patient was utilized; their choices, their answers to questions, and even their movements within the environments were all noted by the program in order to offer a more individualized and personalized experience each time they entered the Bubble.

The VR design was created as a safe and tranquil space in which to fully embrace the VR experience. Each environment allowed Luna (our helper) to engage with the patient using CBT and mindfulness. Luna spoke with the patients, gently asking them questions regarding their current needs; allowing them greater control of their environments; and letting them decide how long to stay, what to do there, and where to go, all in an effort to increase the calming effect of the intervention. 

Though CBT and mindfulness techniques were utilized, the authors believe that the VR environment was a critical element of the therapeutic effect as the participant was able to detach and become fully immersed in virtual reality. See Appendix B.

## 2. Methods

### 2.1. Procedure 

The study took place at Oncology & Hematology Specialists, P.A., a private clinic located in Mountain Lakes, NJ, USA. After IRB approval, women aged 18–60 with breast or ovarian cancer were recruited for the study via e-mail or through their personal physician. After we obtained written informed consent, a team clinician conducted a clinical interview to confirm that all inclusion criteria were met. Additionally, the clinician assessed the patient’s initial motivation and general suitability for the intervention (e.g., the patient had a smartphone and was able to comply with study procedures).

Inclusion criteria for the study included:

1. Females between the ages of 18 and 60 with a diagnosis of breast or ovarian cancer who were receiving standard-of-care chemotherapy, endocrine therapy, or estrogen-blocking anticancer treatment.

2. Patients reported experiencing hot flashes for at least thirty days prior to the start of the study, as well as daily hot flashes during the week prior to enrollment.

Exclusion criteria included: chronic migraine headaches, seizure disorders, serious vestibular disorders, serious psychiatric disorders (including but not limited to active suicidality, psychosis, or bipolar disorder), as well as pathological vertigo, pregnancy, drug use including medical marijuana, or currently participating in other forms of psychotherapy. 

All eligible women then participated in a comprehensive presentation of Bubble, along with a short explanation of VR, the rationale for using it to assist with side effects of cancer treatment, and the evidence supporting its use. Patients were asked to complete a self-report survey that included a number of validated questionnaires (see below) and that took between 20 and 40 min on a clinic computer. Figure 1 presents a flow chart of the study.

### 2.2. Participants

Table 1 presents the main background characteristics of the sample. Forty-two women with breast or ovarian cancer participated in the study. The mean age was 47 years (range 25–60 years). Patients were diagnosed with different cancer subtypes, in a variety of stages ranging from 1 to 4. They suffered from hot flashes at different frequencies.

### 2.3. Using VR

Patients initially downloaded two mobile apps, Bubble and Luna (see Appendix A). 

Bubble is a virtual reality (VR) mobile application that offers a psychological intervention based on elements from both CBT (addressing one’s thinking patterns) and mindfulness-based stress reduction (focusing attention on one’s thoughts, emotions, and sensations at the present moment). The intervention takes place in a VR environment in a winter wonderland setting called Frosty. Frosty provides both a virtual reality winter experience and guided meditation. The cold, winter-like experience is designed to help patients calm down and feel cooler. Luna is an avatar (a graphic figure) within the Bubble application. Luna functions as a virtual coach and leads the therapy. The texts are read to patients in Luna’s voice throughout the intervention. Luna is also a separate app that can be downloaded that offers daily reminders to patients to use Bubble and requests feedback on the use of Bubble from patients twice a week. 

Patients were instructed on how to use Bubble at home twice a day (morning and evening), as well as specifically when experiencing a hot flash. 

For this study, patients downloaded the apps to their mobile phones, placed the mobile phone inside designated VR goggles, and then wore a combined device on their head. Once patients entered the VR app, they initially were presented with a preparatory CBT session (4–5 min) in which they were asked to observe their thinking patterns and sensations. This phase was meant to increase one’s attention to one’s feelings, cognitions, and physical sensations before one enters the VR intervention itself. Following the completion of the coaching session, patients transitioned into the Frosty environment, which simulates a cold, winter-like experience, where they went through a guided meditation in Luna’s voice. At the end of each session, the patient returned to the first screen (i.e., where the preparatory phase was conducted) with the intent of “collecting” their thoughts and sensations (again, 4–5 min). All psychological texts within Bubble were written in collaboration with CBT and mindfulness experts.

Participants were asked to use the application for 24 consecutive days, following which they were asked to complete the post-intervention questionnaires. 

## 3. Measures

Demographic and medical information: Demographic information included birth date, ethnicity, education, marital status, employment status, and household income. Date of diagnosis, stage of disease, and types and dates of cancer treatments were also obtained. 

The Hot Flash Related Daily Interference Scale [22] is a 10-item scale measuring the degree to which hot flashes interfere with nine daily activities (work, social activities such as family time, leisure activities such as relaxing and hobbies, sleep, concentration, etc.) as well as overall quality of life. Participants rate the degree to which hot flashes interfered in relation to each item during the previous week using a scale of 0 (did not interfere) to 10 (completely interfered) points. A total score is calculated by adding up the scores of all items (reliability: 10 items; α = 0.95). 

The Hot Flash Rating Scale [23] measures the frequency and intensity of hot flashes and night sweats per day and per week. It also estimates the chronicity and average duration of hot flashes, as well as provides rating on three separate 10-point scales (1 being the least and 10 being the most) for the amount of distress and the degree to which hot flashes were a problem in patients’ lives (reliability: 7 items; α = 0.77). 

The Perceived Stress Scale [24] includes 10 items assessing the degree to which situations in one’s life during the past week were appraised as unpredictable, uncontrollable, and overwhelming, from 0 (never) to 4 (very often). Scores were summed for analyses, and established cutoff scores were also employed. The Perceived Stress Scale was found to be highly reliable (reliability: 10 items; α = 0.85).

The Kessler Psychological Distress Scale [25] is intended to yield a global measure of distress based on questions about anxiety and depressive symptoms. This measure was designed for use in the general population; however, it can also serve as a useful clinical tool. The K-10 comprises 10 questions that are answered using a 5-point scale (where 5 = all of the time and 1 = none of the time). For all questions, the client circles the answer truest for them in the past four weeks. Scores are then summed, with the maximum score of 50 indicating severe distress and the minimum score of 10 indicating no distress (reliability: 10 items; α = 0.89).

The Brief Illness Perception Questionnaire [26] is designed to provide the simple and rapid assessment of illness perceptions using items on a scale from 1–10 including perceived consequences, timeline (acute vs. chronic), amount of perceived personal control, treatment control, identity (symptoms), concern about the illness, coherence of the illness, and emotional representation of the illness. The B-IPQ has demonstrated good psychometric properties, including concurrent, predictive, and discriminant validity for a variety of illnesses and disorders (reliability: 10 items; α = 0.80).

The Pittsburgh Sleep Quality Index [27] is a self-rated questionnaire assessing sleep quality and disturbances over a 1-month period. Nineteen individual items generate seven component scores: subjective sleep quality, sleep latency, sleep duration, habitual sleep efficiency, sleep disturbances, use of sleeping medication, and daytime dysfunction. The sum of the scores for these seven components yields one global score (reliability: 7 items; α = 0.58).

The World Health Organization Quality of Life Scale-Brief Version [28] includes 26 items covering four QOL domains (psychological, physical, social, and environmental). Items are rated from 1 (not at all) to 5 (an extreme amount). Scores for each subscale were summed for analyses (reliability: 26 items; α = 0.87).

## 4. Results

In order to examine changes in various outcome measures during treatment (i.e., pre- to posttreatment), we conducted a series of paired t-tests. Due to multiple comparisons, Bonferroni corrections were applied. Cohen’s [29] effect size was calculated separately for each effect, with effect size of 0.3 representing a small effect, 0.5—medium effect, and 0.8 or above—large effect. 

### 4.1. Pre- to Posttreatment Changes in Hot Flashes and Illness Perception

Our primary outcome measures were those related to hot flashes, as the direct aim of Frosty was to alleviate these physical symptoms. Paired t-tests showed a significant reduction in the overall score on the HFRDIS, i.e., participants reported more disturbance in their life due to hot flashes pretreatment (M = 59.51, SD = 28.27) than posttreatment (M = 48.05, SD = 26.18), t(36) = 2.742, *p* < 0.01, Cohen’s d = 0.42. Furthermore, women in our sample reported a significant reduction in the daily frequency of hot flashes from pre- (M = 11.31, SD = 8.54) to posttreatment (M = 6.83, SD = 4.94), t(28) = 3.367, *p* < 0.01, Cohen’s d = 0.60. Finally, participants reported more positive cancer perceptions (measured as the b-IPQ score) posttreatment (M = 33.51, SD = 4.5) compared with pretreatment (M = 41.64, SD = 14.64), t(36) = 3.8, *p* < 0.01, Cohen’s d = 0.62.

### 4.2. Pre- to Posttreatment Changes in Psychological Distress and Quality of Life

General psychological distress: A paired-samples t-test was conducted to examine the changes in mood and anxiety symptoms, as measured by the K-10 scale. The results indicated a significant reduction in K-10 scores from pre- (M = 20.81, SD = 6.71) to posttreatment (M = 17.27, SD = 5.64), t(36) = 3.00, *p* < 0.01, Cohen’s d = 0.57.

An additional analysis was conducted, this time assessing the change in K-10 levels, according to the scale’s established norms: “likely to be well” (score of under 20), mild symptoms (20–24), moderate symptoms (25–29), and severe symptoms (30 and above). In order to assess the changes in the percentages of patients meeting each K-10 category from pre- to posttreatment, we employed the marginal homogeneity test. Figure 2 presents the distribution of patients before and after treatment.

As can be seen in Figure 2, the rate of participants who were defined as doing well increased over 20% from pre- to posttreatment. Importantly, the percentages of women who experienced moderate or severe psychological symptoms were 6 times higher and 2.5 times higher, respectively, before than after using Frosty (Mean MH statistic = 37.5, *p* < 0.05). 

Quality of life: Of the four quality of life domains included in the WHOQOL-BREF measure, two showed significant increases from pre- to posttreatment: physical quality of life and psychological quality of life: patients reported significantly lower physical quality of life before (M = 11.93, SD = 2) than after (M = 15.36, SD = 2.47) using Frosty, t(36) = −11.23, *p* < 0.001, Cohen’s *d* = −1.49. A significant change was also found in psychological quality of life, with lower scores reported before (M = 13.66, SD = 2.65) than after (M = 14.47, SD = 2.67) treatment, t(36) = −2.97, *p* < 0.01, Cohen’s *d* = −0.30. No significant changes were found in environmental QOL and social QOL after using Frosty.

Perceived stress: A paired-samples t-test was conducted to examine the level of perceived stress, as measured by the PSS. The results indicated a significant reduction in perceived stress scores from pre- (M = 21.5, SD = 3.90) to posttreatment (M = 19.34, SD = 3.16), t(37) = 2.96, *p* = 0.005, Cohen’s d = −0.61.

### 4.3. Pre- to Posttreatment Change in Sleep Quality

Significant positive changes were found in three PSQI measures. First, participants showed an increase in total PSQI scores (overall sleep quality, comprising all PSQI subscales) from pre- (M = 8.32, SD = 0.3.92) to posttreatment (M = 6.31, SD = 3.74), t (36) = 3.54, *p* < 0.01, Cohen’s *d* = 0.52. Additionally, sleep onset latency significantly changed, with participants reporting that it took them more time to fall asleep before (M = 1.08, SD = 0.953) than after (M = 0.675, SD = 0.78) using Frosty, t(36) = 3.11, *p* < 0.01, Cohen’s *d* = 0.46. Finally, participants’ subjective evaluations of their sleep quality was lower before (M = 1.07, SD = 0.72) compared with after (M = 0.50, SD = 0.74) using Frosty, t (27) = 3.151, *p* > 0.01, Cohen’s *d* = 0.81. 

### 4.4. Overall Assessment of Patient Satisfaction with Frosty

In order to assess patients’ general satisfaction with Frosty, they were presented with several single-item questions. A series of binomial tests was conducted in order to examine whether a proportion from each single dichotomous variable was equal to a presumed population value. In this case, the random value was set at 50%, and thus, the test was able to show whether participants’ answers significantly differed from the expected 50–50 distribution. Figure 3 and Figure 4 present the distributions of patients according to their answers on several general assessment items. 

Patients reported overall high satisfaction with Frosty and the VR experience: 70% (N = 21) of patients reported that Frosty assisted them in getting back to their daily routine (*p* < 0.05) and 97% (N = 30) that they enjoyed the Frosty experience (*p* < 0.001). Furthermore, 84% (N = 26) of patients reported their desire to use Frosty on a daily basis if it was offered to them (*p* < 0.001), and 97% (N = 30) noted that they will recommend it to other patients (*p* < 0.001). Additional survey questions referenced the more technical aspects of Frosty, and participants indicated very high satisfaction. The vast majority of patients (90% N = 28) reported having liked Luna (*p* < 0.001), the Frosty virtual guide, and all patients (100% N = 31) noted that they liked her voice (both *p* < 0.001). Finally, 94% (N = 29) of patients reported having liked the Frosty home experience (*p* < 0.001).

### 4.5. What Works for Whom? The Role of Age and Psychological Distress at Baseline

Following our analysis of the main effects of treatment (i.e., pre- to posttreatment changes in various outcomes), we set out to examine whether women in our sample differentially benefitted from treatment as a function of their individual characteristics. More specifically, we examined whether the woman’s age and initial psychological state would moderate the treatment’s effectiveness. We did so by conducting a series of repeated-measures analyses of variance (ANOVAs) with age/T1 K-10 scores as the between-subject independent variables, time as the within-subject independent variable, and several outcome measures as the dependent variables. Significant interactions were found only for sleep quality measures, and these results will be presented here. 

Global sleep quality: Age was found to moderate the effect of Frosty on global sleep quality. For this analysis, the age threshold yielding a significant result was 45 years old (F(1,35) = 4.31, *p* < 0.05, partial ƞ^2^ = 0.109). Figure 5 presents these results (higher scores indicate more sleep problems).

A follow-up simple-effects analysis showed that while the pre- to post-VR difference in sleep quality was significant for women under the age of 45 (*i-j* = 3.33, *s.e.* = 0.66, *p* < 0.001), no difference was found for those over 45 (*i-j* = 1.07, *s.e.* = 0.78, *p* = 0.19).

Sleep latency: Women’s initial level of psychopathology (K-10) was also found to moderate the effect of treatment on sleep latency. Specifically, we found a significant interaction between pretreatment K-10 scores and time (pre- to posttreatment) when sleep latency was the dependent variable (F (1,35) = 7.34, *p* = 0.01, partial ƞ^2^ = 0.17). For this analysis, K-10 scores were dichotomously categorized into low (mild distress and below) or high (moderate to severe) by combining categories according to the scale norms presented above. These results are presented in Figure 6.

A follow-up simple-effects analysis showed that while the pre- to post-VR difference in sleep latency was significant for women who reported high levels of psychopathology before treatment (*i-j* = 0.91, *s.e.* = 0.22, *p* < 0.001), no difference was found for those who reported low psychopathology (*i-j* = 0.19, *s.e.* = 0.14, *p* = 0.19). Thus, while women with elevated psychopathology levels reported that the amount of time it took them to fall asleep grew shorter than from before using Frosty, the improvement among women with low psychopathology was not statistically significant. 

## 5. Discussion

This pilot study aimed to examine whether the AI–VR Bubble app significantly reduced the intensity and/or frequency of hot flashes and improved psychological well-being and sleep in women with breast and/or ovarian cancer. 

Overall, our results provide preliminary evidence for the effectiveness of VR as a potent therapeutic tool among breast and ovarian cancer patients. The Bubble app resulted in reduced intensity and frequency of hot flashes, psychological distress (anxiety, perceives stress and mood symptoms), and insomnia, as well as in improved physical and psychological quality of life. The reduction in hot flashes was found in two main areas, disturbance to one’s life and frequency, indicating VR’s unique potential in alleviating this disturbing symptom. These results are in line with other studies showing the effectiveness of VR interventions in improving both psychological and physical symptoms in various populations [30] including cancer patients [31]. Moreover, after they used Bubble, the women showed more positive perceptions of their illness, an issue known to be critical when dealing with chronic illnesses such as cancer [32]. Interestingly, while the Frosty module of our VR product directly targeted hot flashes, our findings indicate its effectiveness with psychological factors as well. This further elucidates the mind–body connection and its veracity in chronic disorders. Physical symptoms do not exist on their own but rather are deeply connected and influenced by emotional, cognitive, and behavioral well-being [33]. 

The effectiveness of VR can be attributed to several factors, most notably the vividness of experience provided by this tool [34], which enables patients to not only think about their symptoms or employ traditional exposure techniques commonly used in psychotherapy but actually to work with a very tangible, physical experience and witness its change moment to moment. Importantly, patients expressed very high satisfaction (97%) with the VR product, and the vast majority stated that they would use Frosty daily if it were available to them. These results reveal the high applicability of VR products among cancer patients. As VR technology becomes gradually more available, and as prices become more affordable, VR may become a treatment of choice for a variety of medical care providers. 

In this study, we also assessed a rather neglected question in the VR literature pertaining to the heterogeneity of treatment response. More specifically, we examined whether patients’ age, as well as their pretreatment psychological well-being, were associated with the degree to which they had benefited from VR. Interestingly, the effect of Frosty seemed to be age related, with a significant improvement in sleep quality among younger patients. This may be attributed to several factors, including younger patients’ increased familiarity and comfort with advanced technology compared with their older counterparts. In addition, younger age may be associated with lower cancer chronicity, which can in turn impede recovery. As for initial level of psychopathology, only women with pre-VR elevated psychopathology levels reported that the amount of time it took them to fall asleep grew shorter from before to after using Frosty. Sleep and psychopathology were found to be interrelated in numerous studies [35]. Thus, it may be that sleep disturbances were particularly relevant among those patients who had previous emotional difficulties and that these subjects particularly benefited from this aspect of VR treatment and needed it the most. The question of what works for whom [36] in both medical and psychological practice is gradually becoming the center of attention in intervention studies. Numerous other factors (e.g., personality traits, biological markers, cognitive styles) are likely to play a role in VR treatment response and should be the focus of future studies. 

The most significant limitations of this study are the lack of a control group, a relatively modest sample size, and the use of self-report measures as well as a short intervention with no follow-up assessment. As VR is an emerging therapeutic field, there is still much room for larger studies based on more sophisticated methodologies, most notably rigorous randomized controlled clinical trials (for specific methodological observations, see [37].

Nonetheless, this pilot study provides highly encouraging preliminary evidence for the potential and promise of VR as a therapeutic intervention for hot flashes and accompanying psychological and physical difficulties. VR is easy to use, both at the clinic and at one’s own home. It provides a vivid environment that enables the patient to quickly develop real-life skills and coping strategies, which are highly important when dealing with serious illness. Both physicians and mental health professionals are encouraged to further explore and utilize this unique tool for the treatment of cancer patients, as well as other vulnerable populations.

## Figures and Tables

**Figure 1 healthcare-10-02261-f001:**
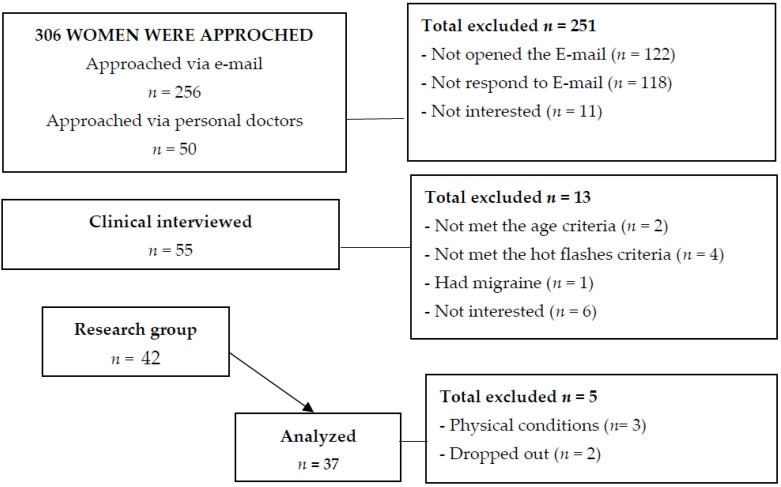
Study flow chart.

**Figure 2 healthcare-10-02261-f002:**
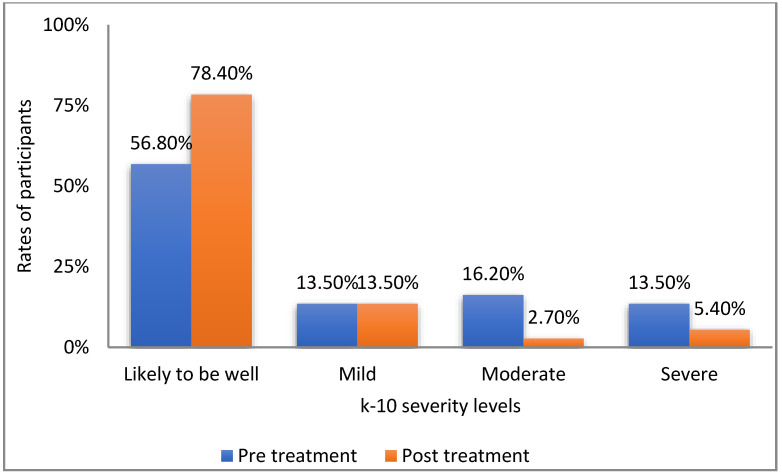
Rates of participants according to k-10 severity levels.

**Figure 3 healthcare-10-02261-f003:**
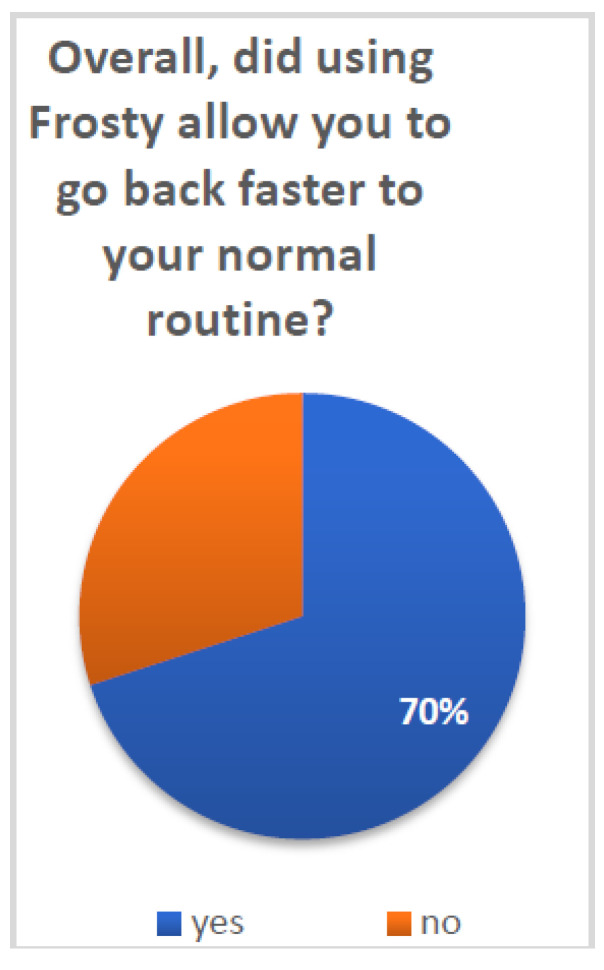
Going back to routine after using Frosty.

**Figure 4 healthcare-10-02261-f004:**
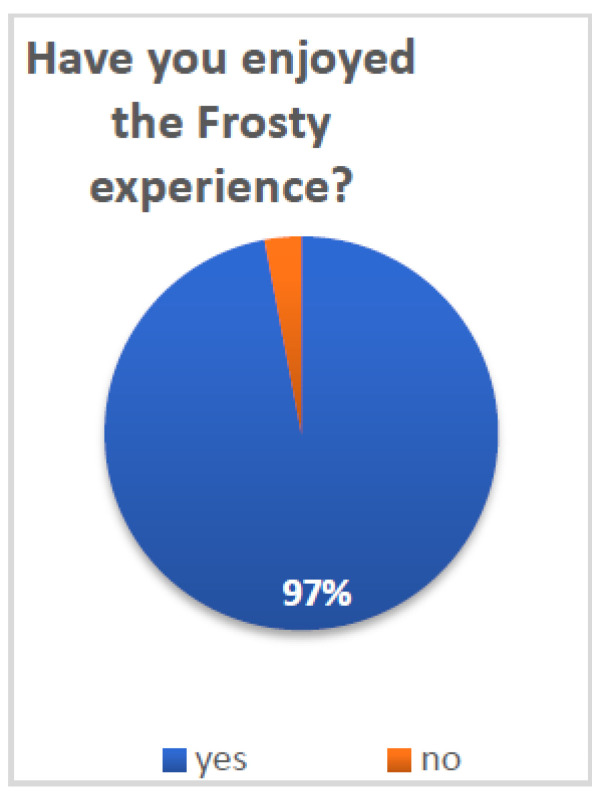
Enjoying the Frosty experience.

**Figure 5 healthcare-10-02261-f005:**
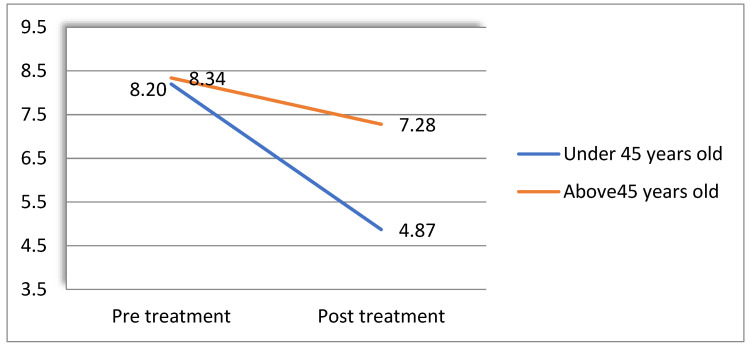
Age moderates the effect of treatment of global sleep quality.

**Figure 6 healthcare-10-02261-f006:**
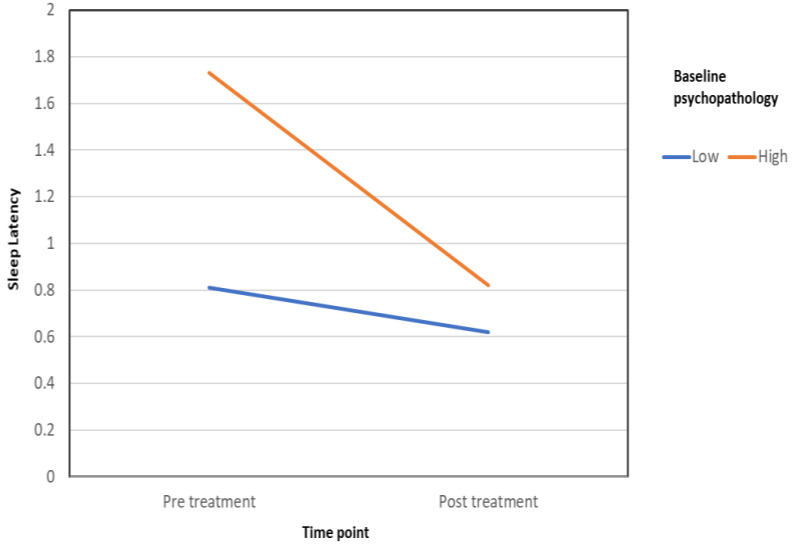
Baseline psychopathology level moderates the effect of treatment on sleep latency.

**Table 1 healthcare-10-02261-t001:** Sample characteristics.

Variable	N = 42	M	SD	%
Age		46.78	6.8	
25–34	2			4.8
35–44	12			28.5
45–54	21			50
55–60	7			16.7
Marital status				
Married	27			64.3
Single	4			9.5
Divorced	4			9.5
Separated	2			4.8
In a relationship	5			11.9
Children				
Yes	32			76.2
No	10			23.8
Employment				
Not employed	10			23.8
Part time	11			26.2
Full time	21			50
Income				
Under average	7			16.7
Average	9			21.4
Above average	23			54.8
Chose not to answer	3			7.1
Time since diagnosis				
1–2 years ago	20			69
3–4 years ago	10			23.8
5–6 years ago	3			7.2

## Data Availability

The data is not accessible at this time.

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
