# Peer review of "Virtual Reality Combined with Artificial Intelligence (VR-AI) Reduces Hot Flashes and Improves Psychological Well-Being in Women with Breast and Ovarian Cancer: A Pilot Study"

_healthcare, 2022, doi:10.3390/healthcare10112261_

Round 1

Reviewer 1 Report

The authors present an interesting use case of VR to reduce hot flashes in women with breast and ovarian cancer. The manuscript is well written and clearly structured. I am glad to see that the authors recognise the need to examine factors that influence treatment outcome in the context of VR treatments, with respect to personalised medicine. However, some aspects of this research and manuscript may require some more attention.

I found the mentioning of AI in the title and background somewhat misleading as the manuscript fails to provide any information on which AI technology has been used in the proposed treatment and for which purpose. The methods section could be extended to include a section on the use of AI. It should be further discussed what the advantage and impact of this technology was on the study.

Although the focus of this work is the development of a VR treatment and its evaluation in a small study, the description of this VR treatment is quite limited and lacking significant detail for the reader. A more thorough description of the VR environments accompanied by graphical material, and of the integration of the treatment approach (guided meditation) in the VR environment would be interesting. Furthermore, be careful with the interpretation of VR as a treatment method in itself, rather than an immersive medium to provide treatment with. VR does not define the treatment, but the therapeutic methods that use the technology does. VR can simply provide a context in which these methods are more effective, e.g. by incorporating story telling in VR, providing immersion and blocking the outside world. Especially consider this when discussing related work.

Although the results are clearly described and supported by sound statistical analysis, I am not convinced by the conclusions of the authors that the results show the effect of VR in this treatment, nor the effectiveness of VR. This study investigates the effect of the treatment approaches (CBT, mindfulness) in VR as a whole instead of the effect of VR on the treatment outcome. 

As the authors already rightly mention, the lack of a control group or a extended within-subject baseline periode before the intervention is a sever limitation of this work. This further supports my concerns about the conclusions on the effect of VR in this study. Perhaps a follow-up with the participating patient could still provide some interesting insights.

Finally, be careful of the term effectiveness. Clinical randomised controlled trials test efficacy not effectiveness. 

Author Response

Dear reviewer

Thanks for your comments and thorough review 

The manuscript has been updated to include a detailed explanation of how bubble is AI

Thanks

Joe Feuerstein MD

Reviewer 2 Report

The paper title mention the word Artificial Intelligence, however, in the manuscript does not appear any information about the algorithm used. The authors does not mention if they use machine learning, deep learning, nature-inspired or other type of artificial intelligence.

I think the results are not objective because depends on the patient perception.

Author Response

(The authors gave the same response as above.)

Reviewer 3 Report

(1) The manuscript needs English proofreading. (2) What is the limitation of the proposed approach?

Author Response

Dear reviewer

Thanks for your comments and thorough review 

The manuscript has been updated to include a detailed explanation of how bubble is AI, has undergone a review for any grammatical errors and the discussion section does note the obvious limitation of this study; a lack of a control group. 

Thanks

Joe Feuerstein MD

Round 2

Reviewer 1 Report

The additions and changes to the manuscript are rather limited. Although a bit more light has been shed on the use of AI in this study it still remains vague . I'm interested in the AI algorithms that have been used and details as to how the VR experience has been personalised. Which effects did the patients choices, answers and movement have on this experience?

I understand that the authors believe the use of VR has a significant therapeutic effect, however, I still don't see this hypothesis supported by the results of this study. 

Author Response

Dear reviewer 

Regarding the request for more information on the AI algorithm, please see the final manuscript that was edited to give much more detail about the AI technology utilized in the study 

We believe that this study showed a number of positive effects for VR therapy, in several important life domains – hot flashes, psychopathology, quality of life with large effect sizes, further supporting the effectiveness of this intervention. The reviewer points out that he/she cannot be certain that it was VR in itself that yielded the effects, and this is of course always true. Nonetheless, we have good reason to believe that VR played a big part in the success of this interventions, for several reasons:

  1. As noted, there were positive outcomes in various domains, thus excluding the possibility of an arbitrary, isolated effect.
  2. Hot flashes were the main target of this intervention, and the VR was designed and programmed specifically for this in mind. We posit that one of the main causes of the hot flashes response to this intervention due to the sensory and physical experience created by the VR.
  3. We deliberately included questions assessing satisfaction with VR specifically. As noted (see page 12) Patients reported overall high satisfaction with Frosty and the VR experience. 70% (N = 21) of patients reported that Frosty assisted them in getting back to their daily routine ( p<0.05), and 97% (N = 30) that they enjoyed the Frosty experience (p <0.001). Furthermore, 84% (N = 26) of patients reported their desire to use Frosty on a daily basis, if offered to them (p <.001), and 97% (N = 30) have noted that they will recommend it to other patients ( p< 0.001). Additional survey questions referenced the more technical aspects of Frosty and participants indicated very high satisfaction. Thus, the vast majority of patients (90% N = 28) reported having liked Luna (p <0.001), the Frosty virtual guide, and all patients (100% N = 31) noted that they liked her voice (both p <0.001). Finally, 94% (N = 29) of patients reported having liked the Frosty home experience (p <0.001).

Thus, we did not rely solely on outcomes, but also assessed aspects pertaining to the device and technology themselves.

Overall, we believe that considering all of the above we can attribute at least some of the results in this study to VR. However, due to the reviewer’s comment, we have now added several sentences acknowledging the fact that other factors other than VR may have played a role (including mindfulness, CBT aspects, or other psychological factors which we did not assess – please see page 15 for these new additions). This, we believe, presents a more balanced picture of our findings.

Reviewer 2 Report

The information about the type of artificial intelligence is null, authors did not mention what kind of algorithm they used for experiments. For publication, is mandatory to include a detail explanation of the artificial intelligence algorithm

Author Response

Dear reviewer

Please see the new final manuscript that specifically details the AI algorithm 

Thanks